# Determining the drivers of global innovation under COVID-19: An FSQCA approach

**Zhenxing Gong**[1,2]*, **Yue Wang**[1], **Miaomiao Li**[3]

1 Business School of Liaocheng University, Liaocheng, China, 2 University of Wisconsin-Madison, Madison, WI, United Stats of America, 3 School of Economics and Management, Beijing Information Science and Technology University, Beijing, China

* zxgong118@163.com

## Abstract

During the COVID-19 epidemic, national innovation faced the challenges of high-risk research and development and intensified trade competition. How to allocate resources reasonably to promote national innovation has become a problem that must be solved. Based on the global innovation index (GII) framework, this study analyzes the influence of national innovation input elements (such as human capital resources, infrastructure, business maturity, etc.) on innovation output from the perspective of configuration, combining with the necessary condition analysis (NCA) and fuzzy set/qualitative comparative analysis (FSQCA). The research results show that:(1) A single innovation input constitutes the necessary condition and serves as a bottleneck for high innovation output;(2) ITT, HCR, IFT, MS and BS are all "multiple concurrent" and form different configurations, namely, two high-innovation and four nonhigh innovation configurations, that drive national innovation governance is characterized by "different roads leading to the same goals." (3) As innovation is limited by the income levels of various countries, there are obvious differences in innovation drive paths between high- and low-income countries. Moreover, the configuration of asymmetric relationships with low-innovation output that occurs in high-income countries has unique characteristics. In this study, the influence of the coupling of national innovation input elements on innovation output is explored.

## 1. Introduction

In 2021, an epidemic spread worldwide and altered international social and economic structures. The World Health Organization named it COVID-19. How to manage the virus and when and how to open the economy again have become global issues [1]. In response, governments were faced with the challenges of high-risk research and development (R&D), insufficient investment in innovation caused by the resulting economic downturn, and intensified trade competition due to intercountry control [2]. To address these challenges, it is necessary to rationally allocate resources to promote innovation according to their own national conditions [3].

Different support systems and cooperative organizations related to industries form an innovative ecosystem that is both interdependent and symbiotic [4]. The relationship between

**Data Availability Statement:** All relevant data are within the paper and its Supporting Information files.

**Funding:** This research was supported by grants from the National Natural Science Foundation of

China (71801120,72002016) and the Shandong Provincial Youth Innovation Science and Technology Support Program (2021RW031). The authors in this research have no relevant financial or non-financial interests to disclose and no competing interests to declare that are relevant to the content of this article. All authors certify that they have no affiliations with or involvement in any organization or entity with any financial interest or non-financial interest in the subject matter or materials discussed in this manuscript. Still, the authors have no financial or proprietary interests in any material discussed in this article.

**Competing interests:** The authors have no relevant financial or non-financial interests to disclose. The authors have no competing interests to declare that are relevant to the content of this article. All authors certify that they have no affiliations with or involvement in any organization or entity with any financial interest or non-financial interest in the subject matter or materials discussed in this manuscript. The authors have no financial or proprietary interests in any material discussed in this article. no competing interests to declare that are relevant to the content of this article.

countries is heterogeneously synergetic rather than competitively oppositionist [5]. Actually, in view of the global innovation problem, academic circles have made many beneficial explorations. Many scholars have analyzed mainly the characteristics of innovation and development in a single country or region and made comparisons between these countries or regions [6, 7]. However, most of the above studies explore the independent effects of single variables in accordance with deductive logic, ignoring the combined effect of conditions in the innovation ecosystem. Affected by the epidemic, the global investment policy environment is also full of uncertainties, it is not clear whether the deficiency of a single institutional element hinders innovation and how the conditional configuration systematically affects national innovation. Traditional regression analysis, which simply links income with innovation, may faces practical problems. Since the start of the COVID-19 pandemic, many researches on how to improve global innovation had remained in the stage of theoretical construction, focusing only on the independent role of different factors at the levels of science, technology and organization, which limits the understanding of the synergistic pairing effect of multiple elements of innovation ecosystems to the difference in national innovation output. Based on the configuration perspective, this paper analyzes in-depth whether the configurations and single conditions that drive high innovation cause the bottleneck of national innovation.

The special innovation demand brought about by COVID-19 and the artificially created R&D barriers between countries in response to it have become the constraints of innovation in all countries. Actively responding to the epidemic influence and identifying experiences and lessons from it can provide an empirical basis for countries to complement each other's advantages, innovate collaboratively and overcome difficulties together when a pandemic occurs again. Therefore, against the background of COVID-19, based on the configuration perspective, combined with NCA and FSQCA, this study analyzes the necessary and sufficient causality [8] and identifies the driving path for achieving high innovation.

## 2. Literature review and research framework

### 2.1 Connotation and research progress of the national innovation index

An innovation index is used to quantify a country's capacity for innovation and to evaluate what actions should be taken by a country to encourage innovation and which countries have advantages over other countries in this arena [9]. The United Nations, World Bank, Council of Europe and World Economic Forum (WEF) have all proposed various indices to measure technology and innovation. Cornell University, the INSEAD and the WIPO jointly published the GII 2021 edition, in which innovation input included ITT (political environment, regulatory environment, and business environment), HCR (education, tertiary education, and R&D), IFT (information and communication technology, general IFT, and ecological sustainability), MS (credit, investment, trade, diversification, and market size), and BS (knowledge workers, innovation linkages, and knowledge absorption). Innovation output includes two dimensions, namely, knowledge and technical output (knowledge creation, influence and dissemination) and creative output (intangible assets, creative products and services, and network creativity), in which each subdimension consists of different indicators, resulting in a total of 81 indicators (Cornell University, INSEAD, & WIPO, 2021). Previous studies have considered all of the aspects of national innovation capability measurement, and the innovation index has been deemed appropriate for representing the technological progress and innovation capability performance of different countries [10]. The reason that this study uses GII data is that in the past, innovation indices were mostly selected from the data of specific organizations by selecting some subjectively important innovation factors, while GII includes date beyond only that of other organizations, particularly WIPO date, when considering intangible assets

(Cornell University, INSEAD, & WIPO, 2021). Many studies have confirmed that the national innovation capability index derived from the GII is very appropriately representative [3, 11].

Previous studies have focused on the GII from the following three aspects. First, the characteristics of innovation in a certain country or region were determined and compared. Some scholars have analyzed the characteristics of innovation input and output in a single country [6, 7]. For example, Al Qallab et al.(2018) analyzed the innovation index of countries in the Arab region and pointed out that although the innovation index of countries in the Arab region improved from 2008 to 2016, the GII of Arab countries was lower than that of the entire world. A possible reason for this finding is that in 2013–2016, the global information IFT index increased, while the information IFT index of Arab countries decreased during the same period. Vukoszavlyev (2019) found that there are significant differences in GII indicators among continents but there is also strong homogeneity in innovation input and output among low-income countries. Second, a cluster analysis of countries and qualitative policy measures has been proposed. Jankowska et al. (2017) divided global countries into internal homogeneous and external heterogeneous groups, divided the innovation input and output into high and low levels, and adopted two-step clustering technology and descriptive statistical methods to provide targeted policy support for different countries to compensate for the shortcomings of innovation output [12]. Famalika & Sihombing (2021) compared the cluster analysis method, the results of which showed that the variance in the k-median was smaller than that in the k-means, thus the k-median was better than the k-means. Third, the impact of key indicators of innovation input on innovation output and GDP have been underscored. As GII involves many indicators, some researchers have explored the key factors affecting innovation output and the bottleneck factors of high- and low-income countries from the perspective of indicator simplification (Bate et al., 2021). Regardless of whether scholars adopt principal component analysis (AlQaraah, 2022), structural equation modeling (Sohn et al., 2016), or artificial neural network analysis (Pençe et al., 2019), past research has not broken through the symmetrical thinking of traditional linear regression analysis. Although the above research is helpful for guiding the selection of antecedents in this study, previous studies have focused on the positive or negative linear relationship between a certain condition and a specific innovation output. Because of the lack of an interdependent coupling effect between conditions from the perspective of the entire system, it is difficult to directly apply these conditions to the theoretical deduction of complex causality from the perspective of configuration.

## 2.2 Construction of the research framework

Although some studies have begun to analyze the path differences in national innovation output from the perspective of configuration and to compare those of high-income countries to promote high innovation [3, 13], there are still few studies that can explain the differentiation among national innovation paths, especially that during the COVID-19 pandemic. The existing research presents the following shortcomings. First, the promotion of national innovation output involves the interdependence rather than independence of conditions. Second, although the existing research has provided rich explanations for national, regional and even global innovation, these studies face difficulty in providing sufficient theoretical support for the selection of differentiated paths to promote national innovation. Third, in the real world, the governance of national innovation involves the logical relationship between the matched patterns and the results of different conditions. These conditions that lead to high or low innovation output may not be the same. Many studies have shown that the relationship between income and innovation output is not simply proportional [14]. In the past, the complexity of causality between national innovation input and output had not been considered. In view of

the above limitations, this paper introduces the FSQCA method to explore the linkage effect of innovation investment in the context of COVID-19 under the GII framework and reveals the complex relationships among different influencing factors.

First, ITT is a major driving force of economic activities and its optimization results from political, legal system construction and many other internal and external conditions. A stable political environment can reduce the level of business uncertainty and encourage innovative activities. Studies have shown that optimizing the business environment has a positive impact on effective corporate governance, business scope, competitive advantage and innovation ability [15]. A sound business environment system can help new entrants start businesses more easily, solve problems such as bankruptcy and tax payment, reduce the uncertainty of starting a business, and enhance the competitiveness needed for innovation [16].

Second, the level of HCR is the main determinant of innovative output. Innovation is produced by human capital and then transformed into intangible assets, including intellectual assets [17]. Moreover, human capital plays an important role in the generation of innovation through high-quality knowledge [18]. The higher the country's level of human capital is, the stronger its ability to absorb and use high technology, and the more obvious its process of national innovation and progress [19].

Third, IFT, as the basic element of economic development, is the prerequisite for ensuring national development and provisioning hardware guarantee to promote innovation. Communication, transportation and energy IFT promote the exchange of innovative ideas and services [20]. IFT changes affect economic growth and development. IFT, as a basic element of regional development policy strategy, increases the availability of resources, improves productivity, and serves as an important supporting force in promoting the capability of independent innovation and enhancing the level of national competitiveness.

Fourth, MS is composed of market value or value and total investment. A good credit system can effectively protect the rights of both borrowers and lenders and raise funds for innovative activities [21]. MS can influence foreign direct investment, which has a significant impact on productivity growth and knowledge diffusion [22]. The significant impact of knowledge disclosure on market value is closely related to the patent citation index, and innovation depends on technical characteristics, such as patent application and market structure [9].

The fifth aspect is BS. As a development process, the absorption of innovation comprises a continuous feedback loop that includes acquisition, digestion, conversion and utilization [23]. At the national level, the gathering of knowledge workers leads to a region's emphasis on investment in education. The historically positive cooperation among industry, universities, and researchers has induced a number of research institutions to serve enterprises by increase their levels of R&D investment [24]. Making it easier for innovators to profit from their innovations through intellectual property protection. The level of motivation of innovators to generate new knowledge is insufficient, and the patent system provides a way of increasing the level of such motivations.

Therefore, we assume that ITT, HCR, IFT, MS and BS in the GII structure will have different degrees of influence on national innovation.

## 3. Research methods

### 3.1 The essentiality of using FSQCA and NCA

Several tenets suggest moving beyond multiple regression analysis to thinking and using algorithms.

First, based on correlation assumes that the variables are independent, the regression analysis method is suitable for exploring the net effect of a single explanatory variable on the

explained variable. While FSQCA focuses on analyzing the multiple concurrent causal relationships formed by different combinations of causes and conditions, we can reveal the veil of how the causes and conditions interact together to affect the innovation outcomes (Pappas & Woodside,2021). The FSQCA method begins with holism, proceeds through an analysis of the complex causal relationships among many factors, and finally enables a cross-case comparative analysis to explore which configurations of conditional elements cause the expected results and which fail to lead to the expected result and result in other causal complexity problems [25]. National innovation drives the combination of various factors to form different conditional configurations, and the complex influence on high national innovation output represents this type of problem, so the FSQCA method is especially suitable for this research.

Second, unlike correlation analysis, consistency is a test for sufficiency and not a test for sufficiency and necessity and FSQCA applies Boolean algebra to overcome the weakness of regression analysis and keeps the complex variable relationship aligned with further information about the research object, which mitigates the issue of missing variable deviation. The traditional quantitative method seeks to obtain the optimal solution of the result, while the qualitative comparative analysis method believes that the configuration leading to the result has equivalence, that is, the combination of multiple different conditions will produce the same result [26]. FSQCA can identify the sufficiency and necessity of the causes and conditions that cause the results, as well as the complementarity/substitution among different causes and conditions (Woodside,2013), and further deepen the understanding of different types of innovation input to innovation output.

Finally, the necessary condition refers to the condition needed for a specific result to occur. If this condition does not exist, then the corresponding result cannot be produced. NCA quantitatively demonstrates the antecedent level necessary to achieve a certain level of outcome variables by analyzing the effect size and bottleneck level of the antecedents (%) [26].

## 3.2 Method of mixing NCA and QCA

Necessary and sufficient causation are two new explanations for causality, with the former referring to the conditions needed to enable for a particular result [27]. When an antecedent condition does not exist, the result does not occur. Moreover, sufficient conditional causality means that antecedents (or various combinations of antecedents) can fully produce results [25]. Necessary condition analysis is indispensable in configuration analysis. NCA is a sufficient and direct method to identify the necessity of data, and the necessary condition can be expressed as "no X, no Y". In order to make up for the deficiency of QCA analysis method in necessity analysis, NCA analysis method is introduced on the basis of QCA analysis method for further inspection.

First, NCA is a method to analyze data by using R software [28]. And NCA is used in this study to test whether the combination of national innovation drivers is a necessary condition for enabling high-innovation output. If yes, then what is the required level? FSQCA is used to tests the necessary conditions to analyze the robustness of the results.

Second, this study uses the FSQCA method to explore the causal complex mechanism between the combination of national innovation drivers and national innovation output. The FSQCA method begins with holism, proceeds through an analysis of the complex causal relationships among many factors, and finally enables a cross-case comparative analysis to explore which configurations of conditional elements cause the expected results and which fail to lead to the expected result and result in other causal complexity problems [25]. National innovation drives the combination of various factors to form different conditional configurations, and the complex influence on high national innovation output represents this type of problem, so the FSQCA method is especially suitable for this research.

Finally, the FSQCA method effectively supports the implementation of causal complexity analysis [25]. Combining the advantages of qualitative and quantitative research, this QCA method is suitable not only for case studies involving small and medium-sized samples but also for the quantitative analysis of large samples. At the same time, unlike traditional linear regression, FSQCA applies Boolean algebra to overcome the weakness of regression analysis and keeps the complex variable relationship aligned with further information about the research object, which mitigates the issue of missing variable deviation. Therefore, there is no relevant requirement for control variates under the FSQCA method [29].

### 3.3 Data sources

To demonstrate the technological progress and innovation capability of different countries more intuitively, Cornell University, the Institut Européen d'Administration des Affaires (INSEAD) and the World Intellectual Property Organization (WIPO) jointly published the Global Innovation Index (GII) 2021 edition, which provided indicators with which to measure innovation performance and rank the innovation ecosystems of 132 economies around the world, as 2021 was the year in which the whole world had to jointly deal with COVID-19 (Dutta et al., 2021) [30]. The country samples used in this analysis are the same as those referenced in the GII 2021 published in September 2021 (Cornell University, INSEAD, & WIPO, 2021)that reported the innovation of 132 countries worldwide. The choice of each group of countries by income(high-income and low-income countries) is determined using the World Bank income group classification (Cornell University, INSEAD, & WIPO, 2021).

### 3.4 Variable measurement and calibration

Calibrating is a necessary step before analyzing the necessity and sufficiency of antecedent variables and result variables. If a condition always exist when certain results appear, then causal conditions are necessary. If the consistency score is 1, then it indicates that the causal conditions conform to the rules in all cases. According to the standard of Ragin [31], a consistency of greater than 0.8 can be regarded as indicating "almost always necessary".

In this study, direct calibration method is used to calibrate antecedent variables (ITT、 HCR、 IFT、 MS、 BS)and result variables(Innovation). As the GII 2021 is a newly published measurement result, it lacks external and theoretical standards. Referring to previous studies three calibration points are set, namely, full membership, crossover and full non-membership of five conditional variables and one outcome variable, the original values of which are set to cover 95%, 50% and 5% of the data values, respectively (Table 1). The calibration of non-national innovation performance is realized by taking the non-set of high national innovation performance. The calibration results are shown in Table 1.

## 4. Analysis results

### 4.1 Analysis of necessary conditions

The necessary condition refers to the condition needed for a specific result to occur. If this condition does not exist, then the corresponding result cannot be produced. NCA focuses on the factors that can produce or contribute to specific outcomes, and these factors are crucial for organizational decision-making, often being necessary conditions for the organization to produce a specific outcome. The basic logic of necessary (no sufficient) conditions is that if necessary conditions exist, the expected result may not necessarily occur, but if necessary conditions do not exist, the expected result will definitely not occur. NCA quantitatively demonstrates the antecedent level necessary to achieve a certain level of outcome variables by

**Table 1. Descriptive statistics and calibration values of condition and result variables.**

| | Outcome | Conditions | | | | |
|---|---|---|---|---|---|---|
| | InnPer | ITT | HCR | IFT | MS | BS |
| **Country (n = 132)** | | | | | | |
| Min | 5.6 | 27.6 | 7.0 | 17.6 | 23.7 | 8.7 |
| Max | 62 | 95.1 | 67.4 | 64.8 | 84.7 | 68.1 |
| Mean | 25.2 | 64.9 | 32.7 | 41.5 | 47.6 | 29.8 |
| SD | 13.2 | 14.5 | 15.4 | 12.4 | 11.7 | 14.1 |
| **Calibration criterion** | | | | | | |
| 95% | 52.06 | 88.8 | 59.8 | 59.9 | 67.3 | 59.3 |
| 50% | 21.55 | 63.1 | 31.6 | 42.1 | 46.3 | 25.4 |
| 5% | 9.61 | 44.3 | 10.7 | 22.0 | 28.5 | 14.6 |

analyzing the effect size and bottleneck level of antecedents (%). The bottleneck level refers to a certain level at which the maximum observation range of the results is reached and the level value (%) falls within the maximum observation range of antecedents. The closer to 1 that the value is, the larger the effect quantity, and a value of less than 0.1 indicates that the effect quantity is too small [32].

The NCA analysis results are presented in Table 2, including the effects obtained by two different estimation methods, namely, the ceiling regression (CR) and ceiling envelopment (CE) methods. In the NCA method, the necessary conditions must meet two requirements: the effect quantity (d) is not less than 0.1, and the Monte Carlo simulation of permutation tests shows that the effective quantity is significant [32]. On the whole, the effective quantity of all conditions is 0.1 or larger, and the P-value is significant, which shows that these conditions are all necessary conditions for high innovation performance in China.

Table 3 further reports the bottleneck analysis results obtained through the NCA method. The bottleneck level (%) refers to the level value (%) that needs to be met within the maximum observation range of results and antecedents. As shown in Table 4, to achieve 60% innovation performance, the NCA method requires 36.4% national innovation ITT support, 29.4% HCR, 31.3% IFT, 34.4% MS and 38.7% BS.

In this paper, the QCA method is further used to test the necessary conditions. As shown in Table 4, the consistency of need for the necessity of a single condition is generally high (greater

**Table 2. Analysis results of necessary conditions of NCA method.**

| Condition | Way | C-accuracy | Upper limit area (ceiling zone) | Range | Effect size | P-value |
|---|---|---|---|---|---|---|
| ITT | CR | 92.4% | 0.276 | 0.94 | 0.294 | 0.000 |
| | CE | 100.0% | 0.193 | 0.94 | 0.205 | 0.000 |
| HCR | CR | 93.2% | 0.242 | 0.91 | 0.266 | 0.000 |
| | CE | 100.0% | 0.230 | 0.91 | 0.252 | 0.000 |
| IFT | CR | 92.40% | 0.262 | 0.92 | 0.284 | 0.000 |
| | CE | 100.0% | 0.257 | 0.92 | 0.279 | 0.000 |
| MS | CR | 86.4% | 0.262 | 0.94 | 0.278 | 0.000 |
| | CE | 100.0% | 0.241 | 0.94 | 0.256 | 0.000 |
| BS | CR | 92.40% | 0.297 | 0.93 | 0.319 | 0.000 |
| | CE | 100.0% | 0.249 | 0.93 | 0.267 | 0.000 |

Note: a. Membership value of fuzzy set after calibration. b. $0.0 \leq d < 0.1$: "low level"; $0.1 \leq d < 0.3$: "medium level". c. permutation test in NCA analysis (re-pumping times = 10000).

**Table 3. Bottleneck level table of NCA method.**

| Innovation level (%) | Institution | Human capital and research | Infrastructure | Market sophistication | Business Sophistication |
|---|---|---|---|---|---|
| 0 | NN | NN | NN | NN | NN |
| 10 | NN | NN | NN | NN | NN |
| 20 | NN | NN | NN | NN | NN |
| 30 | 3.7 | NN | NN | 2.6 | NN |
| 40 | 14.6 | NN | NN | 13.2 | 11.5 |
| 50 | 25.5 | 14.5 | 15.3 | 23.8 | 25.1 |
| 60 | 36.4 | 29.4 | 31.3 | 34.4 | 38.7 |
| 70 | 47.3 | 44.3 | 47.4 | 45.0 | 52.3 |
| 80 | 58.2 | 59.2 | 63.5 | 55.6 | 66.0 |
| 90 | 69.2 | 74.1 | 79.5 | 66.2 | 79.6 |
| 100 | 80.1 | 89.0 | 95.6 | 76.8 | 93.2 |

Note: a. CR method, NN = unnecessary.

than 0.8). This result is consistent with the NCA result; that is, there are necessary conditions for producing high national innovation performance.

## 4.2 Configuration analysis

In this paper, fsQCA3.0 is used to analyze the configurations that represent different conditions through the same result (high or non-high innovation performance) can be achieved.

**4.2.1 Configuration that produces high national innovation performance.** In this paper, the original consistency threshold is set to 0.8, the proportional reduction in inconsistency (PRI) threshold is set to 0.70, and the case frequency threshold is set to 1. The QCA results are shown in Table 5. Among them, there are two configurations (Configurations 1 and 2) that lead to high innovation. A detailed analysis of each configuration that lead to high national innovation performance is presented below.

First, Configuration 1 uses ITT, HCR, IFT and BS as the core conditions for producing high-innovation output. Second, Configuration 2 underscores the focus of the state on MS, HCR, IFT and BS as the core conditions, which can also produce high national innovation output.

**4.2.2 Configurations that produce non-high innovation performance.** In this study, the six configurations that lead to nonhigh-innovation performance are also examined, among

**Table 4. Necessity test of single condition of QCA method.**

| Conditional variable | Outcome variable | | | |
|---|---|---|---|---|
| | High innovation countries consistency | High innovation countries coverage | Low innovation countries consistency | Low innovation countries coverage |
| ITT | 0.866 | 0.822 | 0.475 | 0.495 |
| ~ ITT | 0.468 | 0.448 | 0.829 | 0.872 |
| HCR | 0.881 | 0.869 | 0.441 | 0.478 |
| ~ HCR | 0.470 | 0.434 | 0.879 | 0.890 |
| IFT | 0.879 | 0.838 | 0.457 | 0.479 |
| ~ IFT | 0.453 | 0.432 | 0.845 | 0.885 |
| MS | 0.840 | 0.792 | 0.523 | 0.542 |
| ~ MS | 0.515 | 0.496 | 0.799 | 0.845 |
| BS | 0.864 | 0.884 | 0.424 | 0.477 |
| ~ BS | 0.489 | 0.436 | 0.897 | 0.879 |

**Table 5. QCA analysis results of high innovation configuration and low innovation configuration.**

|  | High innovation | | Low innovation | | | | | |
|---|---|---|---|---|---|---|---|---|
|  | 1 | 2 | 1 | 2 | 3a | 3b | 4a | 4b |
| ITT | ● |  |  |  | ⊗ |  | ⊗ | ⊗ |
| HCR | • | • | ⊗ | ⊗ |  | ⊗ | ⊗ | ⊗ |
| IFT | • | • |  | ⊗ | ⊗ | ⊗ | ⊗ |  |
| MS |  | • | ⊗ | ⊗ |  |  |  |  |
| BS | • | • | ⊗ |  | ⊗ | ⊗ |  | ⊗ |
| Consistency | 0.97 | 0.98 | 0.96 | 0.96 | 0.96 | 0.96 | 0.94 | 0.96 |
| Raw coverage | 0.74 | 0.71 | 0.71 | 0.68 | 0.71 | 0.74 | 0.71 | 0.72 |
| Unique coverage | 0.05 | 0.02 | 0.02 | 0.01 | 0.03 | 0.02 | 0.01 | 0.01 |
| Consistency of solutions | 0.97 | | 0.92 | | | | | |
| Coverage of solutions | 0.76 | | 0.88 | | | | | |

Note: ● = Core conditions exist; ⊗ = The core condition is missing; ● = Edge conditions exist; ⊗ = The edge condition is missing; Blank means that the condition has no effect on the result.

which Configurations 3a, 3b/4a and 4b constitute second-order equivalent configurations; that is, the core conditions of these configurations are the same (Fiss, 2011). The following is a detailed analysis of each configuration that produces nonhigh-innovation performance:

1. Configuration 1 shows that under the approach that lacks HCR, MS and BS as core conditions, the country produces low-innovation output.

2. Configuration 2 shows that the national innovation output is not high under the approach that lacks HCR, IFT, and MS as the core conditions.

3. Configuration 3 shows that the country produces low-innovation output under the path on which IFT and BS are the core conditions and ITT or HCR are the marginal core conditions.

4. Configuration 4 shows that the national innovation performance is also very low under the path on which ITT and HCR, which are both conducive to innovation, are lacking as core conditions, and either IFT or BS is lacking as a marginal core condition.

## 4.3 Robustness test

This paper tests the robustness of the antecedent configuration of high-innovation performance. First, the threshold of the case number is increased from 1 to 2, and the configurations thus generated are basically the same. Second, the configuration is adjusted to 0.75, the consistency is adjusted to 0.82, and the resulting configuration is basically the same. Therefore, the robustness test confirms the robustness of the results.

## 5. Performance differentiation path of high income with high innovation, low income with high innovation, and high income with low innovation

Due to the influence of the economic development level, geographical location, resource endowment distribution, technical level and other aspects, different countries have obvious heterogeneity in terms of their innovation output. Therefore, according to the GII's classification of high- and low-income countries, this study explores the configuration of high-

**Table 6. Configuration table of high-income high-innovation, low-income high-innovation and high-income low-innovation countries.**

| | high-income with high-innovation | | | low-income with high-innovation | | high-income with low-innovation | | | | |
|---|---|---|---|---|---|---|---|---|---|---|
| | 1 | 2a | 2b | 1 | 2 | 1a | 1b | 1c | 2a | 2b |
| ITT | ● | • | | ● | ● | ⊗ | ⊗ | ⊗ | | ⊗ |
| HCR | | ● | ● | ● | ● | | ⊗ | | ⊗ | ⊗ |
| IFT | ● | | • | • | | ⊗ | ⊗ | ⊗ | ⊗ | |
| MS | ⊗ | ● | ● | | • | | | ⊗ | | ⊗ |
| BS | ● | ● | ● | ● | ● | ⊗ | | | ⊗ | ⊗ |
| consistency | 0.95 | 0.97 | 0.98 | 0.95 | 0.95 | 0.96 | 0.95 | 0.95 | 0.96 | 0.97 |
| Raw coverage | 0.40 | 0.70 | 0.70 | 0.48 | 0.45 | 0.70 | 0.67 | 0.65 | 0.72 | 0.63 |
| Unique coverage | 0.05 | 0.02 | 0.02 | 0.05 | 0.03 | 0.01 | 0.01 | 0.01 | 0.07 | 0.04 |
| Consistency of solutions | 0.95 | | | 0.96 | | 0.93 | | | | |
| Coverage of solutions | 0.77 | | | 0.51 | | 0.84 | | | | |

Note: ● = Core conditions exist; ⊗ = The core condition is missing; • = Edge conditions exist; ⊗ = The edge condition is missing; Blank means that the condition has no effect on the result.

innovation output in countries with different income levels through a comparative analysis of innovation performance in each country and goes on to compare and analyze the asymmetric relationship between income and innovation output to deepen the understanding of the effect of income difference on innovation output. The specific analysis content is presented in Table 6.

As seen in Table 6 above, there are three configurations (paths) in countries with high income and high-innovation performance. In Configuration 1, ITT, IFT and BS appear as the core conditions, while MS is lacking, which indicates that the above core conditions can promote national innovation performance and compensate for the lack of MS. In Configuration 2, HCR, MS and BS are the core conditions, while either ITT or IFT appears as a marginal core condition, which indicates that the above conditions can promote the innovation output of high-income countries.

There are two configurations (paths) in low-income and high-innovation countries. In Configuration 1, ITT, HCR, and BS are the core conditions, and IFT is the marginal core condition. In Configuration 2, ITT, HCR and BS also appear as core conditions, and MS appears as a marginal core condition. Considering the above two configurations, ITT, HCR, and BS are shown to be the key factors for low-income countries for promoting their innovation performance and thus playing a positive role in global innovation.

There are two configurations (paths) in the asymmetric relationship between countries with low income and high innovation and those with high income and low innovation. The configuration shows that even in countries with relatively high-income levels, if there is a lack of ITT, IFT, or BS, then neglecting HCR or lacking MS results in the low-innovation output of the country. This configuration shows that in countries with high-income levels, if there is a lack of HCR and BS, then there is also either a lack of ITT and MS or a lack of IFT, which also leads to low innovation output.

In summary, income level exerts a great impact on national innovation performance, but different countries exhibit obvious differences in terms of their degree of heterogeneity. High-income countries may not produce only high performance, and low-income countries may not produce only low innovation performance. For countries with different income levels, the paths leading to different degrees of innovation performance are different. Therefore,

countries should actively seek ways of promoting their own innovation performance that are in line with their national conditions to improve the level of their national innovation performance.

## 6. Conclusions

Under the objective situation in which COVID-19 continues to impact on people's life and health as well as under global economic development, the rational coordination of resource allocation and the realization of complementary advantages and heterogeneous synergy among countries of the world are difficult problems that hinder the improvement of national innovation efficiency. With the development of the epidemic situation, national innovation faces new opportunities and challenges, and the effective convergence of innovation factors in innovation ecosystems exhibits a new combination. Combining NCA and QCA methods, this study analyzes the relationship between innovation input and output at the national level from the perspective of configuration. First, this work uses the NCA method to find that each innovation input factor under COVID-19 is a necessary condition for producing high innovation and that a single factor can constitute the bottleneck of high-innovation output. Second, two configurations with high-innovation output and four configurations with low-innovation output are found through a consideration of the configuration perspective and application of the QCA method. The two high-innovation configurations reflect multiple ways in which innovation can be realized in different countries. In this study, HCR, IFT and BS play more extensive core roles in the two high-innovation configurations, and most of the four nonhigh-innovation configurations include non-HCR, non-IFT, and non-BS, thus reflecting the impact of innovation on China during the COVID-19 epidemic. According to their current contexts, different countries can compare the similar configurations in the two paths for achieving high-innovation output and handling the relationship between institutions and the market in the direction of institutional and market strengthening logic to achieve high-innovation output. Finally, constrained by the level of economic development in various countries, there are obvious differences in innovation-driven paths between high- and low-income countries. The relationship between income and innovation is neither simple nor linear. The asymmetric relationship between low-innovation output in high-income countries further illustrates that the conditions that cause the heterogeneity in national innovation output levels greatly differ.

## 7. Discussion ang future direction

### 7.1 Theoretical contribution

Compared with previous GII studies, the main theoretical contributions of this paper are presented below

First, based on necessary causality, using the NCA method, it is found that every innovation input element in the GII is a necessary condition for producing high-innovation output and that a single element can constitute a bottleneck for high- innovation output. Although Crespo & Crespo (2016) found that, according to the test methods and standards of Ragin(2000),factors other than MS are almost always necessary, this study applies NCA to test whether a specific innovation input factor is a necessary condition for generating innovation output and, if so, to determine its necessary level. It is found that all elements are necessary conditions for high-innovation output. The inspiration for improving national innovation is that during the COVID-19 epidemic period, the absence of any element in the system hindered a country from promoting high-innovation output through different configurations of innovation input.

Second, this research systematically integrates the elements of innovation investment in the GII and responds to a call for coupling the research on innovation ecological elements. Against

the background of the epidemic, the introduction of the FSQCA method into the research on national innovation governance not only enriches the research methods but also lays the foundation for the formation of the innovation governance decision-making theory of governments in various countries. At present, the existing research is limited mainly to traditional qualitative analysis, and many factors affecting national innovation have been analyzed in the industry in detail, but there are few research results on the linkage effect among various factors [3, 13]. The configuration analysis of national innovation presented in this paper provides a rich and detailed basis and insight to inform the study of the multifactor coupling of national innovation investment. As far as countries worldwide are concerned, the first configuration is in line with the logic of institutional strengthening, which is consistent with the findings of North. Regarding the degree to which the economy encourages innovation, institutions play an important role. The advantages and disadvantages of national institutions can affect knowledge absorption, which is an important aspect of BS. Institutional optimization at the national level can play a supporting role in promoting the innovation capacity of regions and enterprises. The second configuration conforms to the logic of market strengthening, such as that underlying vaccination and vaccine production. Although vaccination has "positive externalities", there is not enough economic demand for vaccines, although this does not mean that vaccines are not needed [33]. Consumers seem to be more willing to pay for treatment than prevention. Furthermore, vaccine developers lack the motivation to invest in the vaccine market.

Third, the configuration analysis of the asymmetric relationship between income and innovation is carried out. According to income, the paths of innovation improvement across countries with different incomes during the epidemic period are discussed. The research shows that there are significant differences in innovation governance paths between high- and low-income countries. For the former, we can make use of the advantages of ITT, IFT, and BS to break the constraints of insufficient MS or to give full play to the advantages of HCR, MS and BS. For the latter, it is necessary to strengthen ITT, HCR and BS and give play to the complementary role of IFT or MS. The asymmetric relationship between high-income countries and low innovation output is caused mainly by the lack of ITT and IFT or of HCR and BS, which further explains the asymmetric causal relationship that leads to the heterogeneity in regard to the level of the national innovation governance.

## 7.2 Practical contribution

Based on the research on the relationship between innovation input and output in different countries, we can provide targeted recommendations. For each country's characteristics, such as ITT, HCR, IFT, MS, and BS indicators, they should flexibly adjust their systems and market logic to optimize the key element combinations in the national innovation system.

Policy makers can start from the following three aspects:

Firstly, each country should focus on promoting the mature development of HCR, MS, and BS. This can be achieved by globally examining or leveraging the institutional advantages of political, regulatory, and commercial environments or the market advantages of credit, investment, and trade. Policy makers need to develop effective policies to optimize the linkage and matching between elements in the innovation ecological environment.

Secondly, we need to actively respond to the bottleneck constraint of MS. For high-income countries, they should further optimize ITT and IFT and strengthen BS; for low-income countries, they should strive to improve ITT, HCR, and BS to achieve the goal of increasing innovation output.

Finally, we need to pay attention to the asymmetric relationship between income and innovation. High-income countries need to formulate reasonable policies to increase innovation

output. Each country should also select appropriate development paths and targeted measures according to its own national conditions and resource endowment characteristics to promote the complementarity and development of global innovation governance capabilities in ITT, IFT, HCR, and BS.

In conclusion, every country's innovation practices need to fully leverage their own conditions and advantages, formulate relevant policies that cater to actual situations, to enhance innovation output and promote the differentiation and complementarity of global innovation governance capabilities.

## 7.3 Limitations and directions for future research

The innovation environment is the sum of internal and external factors that affect the development of innovation activities and the level of innovation performance, including innovation infrastructure, innovation policy system, innovation cultural atmosphere and so on. These factors are closely related and interact with each other, and jointly play a role in innovation activities and innovation performance. The outbreak and spread of COVID-19 epidemic not only affect people's health and lifestyle, but also affects the global innovation environment from many levels and fields, which is reflected in:

Reduce the supply of innovative infrastructure. Threatened by the virus epidemic, people have reduced all kinds of going out and communicating activities, and significantly reduced the scale of all kinds of production and innovation activities. At the same time, in order to curb the spread of the epidemic, local governments at all levels have continuously adopted and strengthened various preventive and control measures to limit crowd gathering and large-scale mobility. In this situation, the construction and operation of all kinds of innovation infrastructure are greatly restricted, and the supply is sharply reduced, which can't meet the needs of all kinds of innovation subjects to carry out innovation activities. Not only is the rate of enterprises returning to work low, universities, research institutes, various intermediaries and platforms can't operate normally, and innovative elements such as related innovative resources, facilities and talents can't play their due roles, which makes the whole innovation system fall into inefficient operation or even stagnation to a great extent.

Increase the cost of innovation policy system. Affected by the COVID-19 epidemic, the government has shifted and adjusted its work focus from the strategic deployment at the national level to the specific path selection of local economic and social development. Many policies and systems to promote and guarantee innovation will inevitably give way to the medical field, which has significantly increased the operating costs of existing innovative policies and systems.

The epidemic situation has an unprecedented impact on innovation, and its negative impact on the entire innovation environment is also enormous. In further research, attention should be paid to addressing the limitations mentioned above. Firstly, this study ignores the influence of international scientific and technological cooperation and knowledge spillover between countries on national innovation capability. The improvement of innovation ability depends not only on how domestic innovation subjects act but also on the interaction between these subjects at the international and national levels. Khedhaouria and Thurik's (2017) suggested that absorbing knowledge from the international market through the openness of the NIS and learning advanced science and technology from foreign countries to promote domestic innovation is essential. Therefore, in future research, we should consider the role of national cooperation factors and further deepen the qualitative analysis. Secondly, this study used only static data and to analyzes the configuration of national innovation during the epidemic period. However, since the application of the QCA method to dynamic time changes

needs to be improved, the sufficient amount of time for collecting the of national innovation input and output across time to promote the rational development of the time-series QCA method remains unclear. Furthermore, it is worth studying how the change track of innovation input affects that of innovation output.

## Supporting information

**S1 Data.**
(CSV)

## Author Contributions

**Writing – original draft:** Zhenxing Gong, Yue Wang, Miaomiao Li.

**Writing – review & editing:** Zhenxing Gong, Miaomiao Li.

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
