## [Decision Letter · Decision Letter 0]

18 Oct 2023

PONE-D-23-25968Configuration Analysis of Driving Country Innovation During COVID-19PLOS ONE

Dear Dr. Wang,

Thank you for submitting your manuscript to PLOS ONE. After careful consideration, we feel that it has merit but does not fully meet PLOS ONE’s publication criteria as it currently stands. Therefore, we invite you to submit a revised version of the manuscript that addresses the points raised during the review process.

We look forward to receiving your revised manuscript.

Kind regards,

Shazia Rehman, Ph.D.

Academic Editor

PLOS ONE

“This work was supported by the National Natural Science Foundation of China [grant numbers71801120], Shandong Provincial Youth Innovation Science and Technology Support Program [grant numbers 2021RW031];Liaocheng City Philosophy and Social Planning Annual key subject(45).”

“The authors have no relevant financial or non-financial interests to disclose.

The authors have no competing interests to declare that are relevant to the content of this article.

All authors certify that they have no affiliations with or involvement in any organization or entity with any financial interest or non-financial interest in the subject matter or materials discussed in this manuscript.

The authors have no financial or proprietary interests in any material discussed in this article.”

Reviewers' comments:

Reviewer's Responses to Questions

**Comments to the Author**

1. Is the manuscript technically sound, and do the data support the conclusions?

Reviewer #1: Partly

Reviewer #2: Partly

2. Has the statistical analysis been performed appropriately and rigorously? 

Reviewer #1: Yes

Reviewer #2: No

3. Have the authors made all data underlying the findings in their manuscript fully available?

Reviewer #1: No

Reviewer #2: No

4. Is the manuscript presented in an intelligible fashion and written in standard English?

Reviewer #1: Yes

Reviewer #2: No

5. Review Comments to the Author

Reviewer #1: Journal: PLOS ONE

Article title: Configuration Analysis of Driving Country Innovation During COVID-19

Manuscript ID: PONE-D-23-25968

General Comments:

This article studies the allocation for the resources according to national conditions to promote national innovation rationally during this epidemic period. The authors use the Global Innovation Index (GII) framework, combining necessary condition analysis (NCA) and fuzzy set/qualitative comparative analysis (fsQCA), and analyzes the linkage effect of various national innovation input elements on innovation output from a configuration point of view. The authors reached the conclusions that two kinds of paths through which to promote national

innovation, which are configurations in which human capital resources (HCR), infrastructure (IFT) and business sophistication (BS) play a core role under the logic of strengthening institution (ITT) and market sophistication (MS) strengthening.

Overview:

The paper is good written and the empirical work does appear to be carefully and correctly done. The research question is quite good and it does make a sufficient new contribution to the literature to be suitable for the PLOS ONE ONLY after MINOR revisons.

In fact, the literature on the configuration analysis of driving country innovation during COVID-19 is quite new.

The contribution of the paper is the analysis of the linkage effect of various national innovation input elements on innovation output from a configuration point of view using the Global Innovation Index (GII) framework, combining necessary condition analysis (NCA) and fuzzy set/qualitative comparative analysis (fsQCA).

The paper is very interesting; and in my view, it needs to be MINOR improved to reach the standard required for publication in this journal.

Specific Comments:

1. Abstract: somehow large, try to be more concise and with the present results from the article (remove the acronyms)

2. Introduction: NOVELTY + results (better explanation); try to reduce it to maximum 2 pages

3. Literature review: better theoretical explanations (the actual explanation is quite general);

4. Section: 2.2 Construction of the research framework (very large 6 pages; try to reduce it to maximum 2 pages)

5. Section 4.1 Analysis of necessary conditions: quite statistical and inexpressive; try to improve the explanations

6. Introduce at least 3-4 figures for better understanding the evolution and the linkage between the variables

7. Discussions: at least 1 page (with the explanations for groups of countries; not general); separate from the conclusions

General considerations: the idea of the article is very good, but the construction of the article is sometimes very technical (statistical) and it is very large. The authors MUST improve the methodology, explanations, and change the article accordingly.

I ONLY recommend this article be published in PLOS ONE after MINOR revisions (methodology, the discussion, reduce the whole article).

Reviewer #2: Dear Author(s),

Very interesting topic and analysis

I have some concerns about the study. They are listed below.

General:

• English is poor. Manuscript should be thoroughly examined and flow of ideas and use of language should be improved to be crystal clear.

• Problem definition should be articulated clearly.

Abstract

• Flow of language is problematic. English should be checked.

• Covid-19 was a pandemic

Introduction

• Para-1-Senrence-1: Too deterministic

• Para-1-Senrence-3: Abruptly innovation becomes a new topic with the provided definition. Narrative flow irritates reader.

• Para-1-Senrence-4: Too deterministic and assertive

• Para-1: It is too long and jumping from one topic to another without proper conjunctions or transition elements to support the flow of narrative.

• Para2-Sentence-3: Topic is abruptly changes. Flow of narrative is problematic.

• Para-2: too long and contains many too assertive arguments when previous works are criticized. Probably the manuscript was developed during the pandemic. The expressions should be checked when “at present” type adverbs are used.

• Para-3 Sentence-1: “Black Swan Incident” simile should be properly explained. Why it cannot be considered “Black Swan Incident”?

• Para-3 Sentence-2: Very strong argument without any evidence. A reference can be useful to increase its persuasiveness.

2. LITERATURE REVIEW AND RESEARCH FRAMEWORK

2.1 Connotation and research progress of the national innovation index

Para-1: Flow of narrative is problematic. Assertive arguments such as “a lot of studies….” with two reference seems inadequate.

2.2 Construction of the research framework.

• There many assertive arguments without any support/evidence from the literature.

• Too long paragraphs with problematic narrative flow.

• Contains clumsily assembled arguments

• Author(s) repeats citing a few certain studies such as Abi Younes et al., (2020) (cited 8 times through the text)

• Covid19 cannot be articulated as “current crises” in Fall 2023. These expressions should be calibrated.

• Johns Hopkins Center for Health Security should be written correctly

3. RESEARCH METHODS

3.1 Method of mixing NCA and QCA.

The necessity of the methods employed in this study should be clearly stated. Reader cannot easily find out this.

3.2 Data sources.

Citations should be checked against references.

3.3 Variable measurement and calibration.

Calibration process seem vague. Clarity is required.

4. ANALYSIS RESULTS

4.1 Analysis of necessary conditions

Analyses conducted to explore the impact of different configuration should be clearly expressed.

5. PERFORMANCE DIFFERENTATION PATH OF HIGH INCOME WITH

HIGH INNOVATION, LOW INCOME WITH HIGH INNOVATION, AND

HIGH INCOME WITH LOW INNOVATION

Full of hypothetical assertive statements without any evidence/reference.

6. CONCLUSIONS, DISCUSSION ANG FUTURE DIRECTION

6.1 Conclusions

Arguments asserting “Causality” seems questionable.

Analyses conducted seems inadequate to generalize any causation relation between variables. It’s adequacy /validity/reliability should be clearly explained with references to convince reader.

6.2 Theoretical contribution

Too deterministic and assertive statements. They should be checked and restated. Not understood clearly.

6.3 Practical contribution

Too deterministic and assertive statements. They should be checked and restated. Not understood clearly.

6.4 Limitations and directions for future research.

We do know why this study ignores international innovation interactions. Is it because of Covid19? This limitation should be explained.

6. PLOS authors have the option to publish the peer review history of their article (what does this mean?). If published, this will include your full peer review and any attached files.

Reviewer #1: No

Reviewer #2: **Yes: **Ufuk Türen

---

## [Author Response · Author response to Decision Letter 0]

26 Oct 2023

Dear Editors and Reviewers:

Thank you to the editor and reviewers for their suggestions and comments. The comments really helped us see the big picture with the introduction.

Reviewer 1

Thanks for your encouragement and suggestions.

Comments

a) Abstract: somehow large, try to be more concise and with the present results from the article (remove the acronyms).

Response: We have revised the abstract according to your requirements to make it more concise and correspond to the current results of the article.

b) Introduction: NOVELTY + results (better explanation); try to reduce it to maximum 2 pages.

Response: We have made significant revisions to the introduction and rewritten it, mainly highlighting the highlights of our article and providing a better explanation of the results, resulting in a more refined content.

c)Literature review: better theoretical explanations (the actual explanation is quite general);

Response: We drew on the advantages of theoretical explanations and provided a more systematic introduction and explanation of practical explanations.

d)Section: 2.2 Construction of the research framework (very large 6 pages; try to reduce it to maximum 2 pages).

Response: We have abbreviated and refined the construction of the research framework according to the comments, reducing the word count by about two pages.

e) Section 4.1 Analysis of necessary conditions: quite statistical and inexpressive; try to improve the explanations.

Response: We have recognized our shortcomings and provided a deeper explanation of the necessary conditions analysis.

f) Discussions: At least 1 page (with the explanations for groups of countries; not general); separate from the conclusions.

Response: We separated the discussion from the conclusion and made it a separate part, providing an explanation of practical contributions, theoretical contributions, and limitations.

Reviewer 2: 

Thanks for your encouragement and suggestions.

Comments

a) English is poor. Manuscript should be thoroughly examined and flow of ideas and use of language should be improved to be crystal clear.

Response: A native English-speaking professor help us to correct English language.

b) Problem definition should be articulated clearly.

Response: We have thoroughly reviewed the manuscript, improved the flow of ideas and the use of language, and made the definition of the problem clear.

c)Abstract: Flow of language is problematic. English should be checked.

Response: A native English-speaking professor help us to correct English language.

d)Covid-19 was a pandemic.

Response: At present, Covid-19 is no longer a pandemic. This is a mistake in our language expression, and we have made modifications to it.

1.Introduction

• Para-1-Senrence-1: Too deterministic.

• Para-1-Senrence-3: Abruptly innovation becomes a new topic with the provided definition. Narrative flow irritates reader.

• Para-1-Senrence-4: Too deterministic and assertive.

• Para-1: It is too long and jumping from one topic to another without proper conjunctions or transition elements to support the flow of narrative.

• Para2-Sentence-3: Topic is abruptly changes. Flow of narrative is problematic.

• Para-2: too long and contains many too assertive arguments when previous works are criticized. Probably the manuscript was developed during the pandemic. The expressions should be checked when “at present” type adverbs are used.

• Para-3 Sentence-1: “Black Swan Incident” simile should be properly explained. Why it cannot be considered “Black Swan Incident”?

• Para-3 Sentence-2: Very strong argument without any evidence. A reference can be useful to increase its persuasiveness.

Response: Due to the excessive number of issues in the introduction section, we have made significant modifications to the section, deleting some sentences that are too deterministic and lack theoretical basis, explaining why innovation has become a new topic, and adding appropriate conjunctions or transitional elements to support the fluency of the narrative. Like following:

By 2021, the COVID-19 pandemic had spread worldwide and altered international social and economic structures. How to manage the virus and when and how to open the economy again have become global issues. In response, governments were faced with the challenges of high-risk research and development (R&D), insufficient investment in innovation caused by the resulting economic downturn, and intensified trade competition due to intercountry control. To address these challenges, it is necessary to rationally allocate resources to promote innovation according to their own national conditions.

Different support systems and cooperative organizations related to industries form an innovative ecosystem that is both interdependent and symbiotic. The relationship between countries is heterogeneously synergetic rather than competitively oppositionist. The GII system vividly reflects the functions of various elements of the innovation ecosystem. It is a useful tool for evaluating innovation performance and can help guide policy-making. Actually, in view of the global innovation problem, academic circles have made many beneficial explorations. Many scholars have analyzed mainly the characteristics of innovation and development in a single country or region and made comparisons between these countries or regions. However, all the above studies explore the independent effects of single variables in accordance with deductive logic, ignoring the combined effect of conditions in the innovation ecosystem. Affected by the epidemic, the global investment policy environment is also full of uncertainties, it is not clear whether the deficiency of a single institutional element hinders innovation and how the conditional configuration systematically affects national innovation. Traditional regression analysis, which simply links income with innovation, faces practical problems. Since the start of the COVID-19 pandemic, research on how to improve global innovation has remained in the stage of theoretical construction, focusing only on the independent role of different factors at the levels of science, technology and organization, which limits the understanding of the synergistic pairing effect of multiple elements of innovation ecosystems to the difference in national innovation output. Based on the configuration perspective, this paper analyzes in-depth whether the configurations and single conditions that drive high innovation cause the bottleneck of national innovation.

The special innovation demand brought about by COVID-19 and the artificially created R&D barriers between countries in response to it have become the constraints of innovation in all countries. Actively responding to the epidemic influence and identifying experiences and lessons from it can provide an empirical basis for countries to complement each other's advantages, innovate collaboratively and overcome difficulties together when a pandemic occurs again. Therefore, against the background of COVID-19, based on the configuration perspective, combined with NCA and FSQCA, this study analyzes the necessary and sufficient causality and identifies the driving path for achieving high innovation. This study is helpful for broadening the perspective of national innovation-related research, deepening the understanding of the driving path and mechanism of national innovation against the background of the pandemic, and promoting the global process of actively responding to a pandemic with innovation.

2. LITERATURE REVIEW AND RESEARCH FRAMEWORK

2.1 Connotation and research progress of the national innovation index

Para-1: Flow of narrative is problematic. Assertive arguments such as “a lot of studies….” with two reference seems inadequate.

2.2 Construction of the research framework.

• There many assertive arguments without any support/evidence from the literature.

• Too long paragraphs with problematic narrative flow.

• Contains clumsily assembled arguments.

• Author(s) repeats citing a few certain studies such as Abi Younes et al., (2020) (cited 8 times through the text).

• Covid19 cannot be articulated as “current crises” in Fall 2023. These expressions should be calibrated.

• Johns Hopkins Center for Health Security should be written correctly.

Response: Firstly, we provided a detailed and detailed description of the literature review and research framework, but found it unnecessary. Therefore, we refined this section. Secondly, there are many arbitrary arguments in the article regarding the connotation and research progress of the National Innovation Index, such as "many studies...", so we have added some references. Finally, we will also replace the arguments for some clumsy combinations.

3. RESEARCH METHODS

3.1 Method of mixing NCA and QCA

The necessity of the methods employed in this study should be clearly stated. Reader cannot easily find out this.

3.2 Data sources.

Citations should be checked against references.

3.3 Variable measurement and calibration.

Calibration process seem vague. Clarity is required.

Response: Firstly, our research method, the hybrid method of NCA and QCA, is a major highlight of our article and should be explained in detail. Its necessity should also be clearly explained to help readers discover this. Secondly, we compared the data sources with the references and verified the citations. Finally, we also explained the calibration process. Like following:

Necessary condition analysis is indispensable in configuration analysis. NCA is a sufficient and direct method to identify the necessity of data, and the necessary condition can be expressed as "no X, no Y". In order to make up for the deficiency of QCA analysis method in necessity analysis, NCA analysis method is introduced on the basis of QCA analysis method for further inspection. 

To demonstrate the technological progress and innovation capability of different countries more intuitively, Cornell University, the Institut Européen d’Administration des Affaires (INSEAD) and the World Intellectual Property Organization (WIPO) jointly published the Global Innovation Index (GII) 2021 edition, which provided indicators with which to measure innovation performance and rank the innovation ecosystems of 132 economies around the world, as 2021 was the year in which the whole world had to jointly deal with COVID-19 (Dutta et al., 2021).

In this study, direct calibration method is used to calibrate antecedent variables( ITT、 HCR、IFT、MS、BS)and result variables(Innovation). As the GII 2021 is a newly published measurement result, it lacks external and theoretical standards. Referring to previous studies three calibration points are set, namely, full membership, crossover and full non-membership of five conditional variables and one outcome variable, the original values of which are set to cover 95%, 50% and 5% of the data values, respectively (Table 1). The calibration of non-national innovation performance is realized by taking the non-set of high national innovation performance. The calibration results are shown in Table 1.

4. ANALYSIS RESULTS

4.1 Analysis of necessary conditions

Analyses conducted to explore the impact of different configuration should be clearly expressed.

Response: The analysis conducted to explore the impact of different configurations was expressed accordingly.

5. PERFORMANCE DIFFERENTATION PATH OF HIGH INCOME WITH

HIGH INNOVATION, LOW INCOME WITH HIGH INNOVATION, AND

HIGH INCOME WITH LOW INNOVATION

Full of hypothetical assertive statements without any evidence/reference.

Response: For arbitrary statements that are full of assumptions, we cite references for reference and argumentation.

6. CONCLUSIONS, DISCUSSION ANG FUTURE DIRECTION

6.1 Conclusions

Arguments asserting “Causality” seems questionable.

Analyses conducted seems inadequate to generalize any causation relation between variables. It’s adequacy /validity/reliability should be clearly explained with references to convince reader.

6.2 Theoretical contribution

Too deterministic and assertive statements. They should be checked and restated. Not understood clearly.

6.3 Practical contribution

Too deterministic and assertive statements. They should be checked and restated. Not understood clearly.

6.4 Limitations and directions for future research.

We do know why this study ignores international innovation interactions. Is it because of Covid19? This limitation should be explained.

Response: Firstly, citing references provides a clear explanation of the sufficiency, validity, and reliability of the conclusion to convince readers. Secondly, we have reviewed and reiterated the overly deterministic and arbitrary statements in our theoretical contributions. Once again, we have reviewed and reiterated the overly deterministic and arbitrary statements in our actual contributions. Finally, regarding limitations and future research directions, we explained the reasons for the decrease in international innovation interaction caused by Covid-19. Like following:

The innovation environment is the sum of internal and external factors that affect the development of innovation activities and the level of innovation performance, including innovation infrastructure, innovation policy system, innovation cultural atmosphere and so on. These factors are closely related and interact with each other, and jointly play a role in innovation activities and innovation performance. The outbreak and spread of COVID-19 epidemic not only affects people's health and lifestyle, but also affects the global innovation environment from many levels and fields, which is reflected in:

Reduce the supply of innovative infrastructure. Threatened by the virus epidemic, people have reduced all kinds of going out and communicating activities, and significantly reduced the scale of all kinds of production and innovation activities. At the same time, in order to curb the spread of the epidemic, local governments at all levels have continuously adopted and strengthened various preventive and control measures to limit crowd gathering and large-scale mobility. In this situation, the construction and operation of all kinds of innovation infrastructure are greatly restricted, and the supply is sharply reduced, which can’t meet the needs of all kinds of innovation subjects to carry out innovation activities. Not only is the rate of enterprises returning to work low, universities, research institutes, various intermediaries and platforms can't operate normally, and innovative elements such as related innovative resources, facilities and talents can't play their due roles, which makes the whole innovation system fall into inefficient operation or even stagnation to a great extent.

Increase the cost of innovation policy system. Affected by the COVID-19 epidemic, the government has shifted and adjusted its work focus from the strategic deployment at the national level to the specific path selection of local economic and social development. Many policies and systems to promote and guarantee innovation will inevitably give way to the medical field, which has significantly increased the operating costs of existing innovative policies and systems.

The epidemic situation has an unprecedented impact on innovation, and its negative impact on the entire innovation environment is also enormous.

Once again, thank you very much for your comments and suggestions.

Thank you and best regards.

Yours sincerely,

Zhenxing Gong and Wag Yue.

---

## [Decision Letter · Decision Letter 1]

15 Nov 2023

PONE-D-23-25968R1Determining the drivers of global innovation under COVID-19:An FSQCA approachPLOS ONE

Dear Dr. Wang,

Thank you for submitting your manuscript to PLOS ONE. After careful consideration, we feel that it has merit but does not fully meet PLOS ONE’s publication criteria as it currently stands. Therefore, we invite you to submit a revised version of the manuscript that addresses the points raised during the review process.

We look forward to receiving your revised manuscript.

Kind regards,

Shazia Rehman, Ph.D.

Academic Editor

PLOS ONE

Reviewers' comments:

Reviewer's Responses to Questions

**Comments to the Author**

1. If the authors have adequately addressed your comments raised in a previous round of review and you feel that this manuscript is now acceptable for publication, you may indicate that here to bypass the “Comments to the Author” section, enter your conflict of interest statement in the “Confidential to Editor” section, and submit your "Accept" recommendation.

Reviewer #1: All comments have been addressed

Reviewer #2: (No Response)

2. Is the manuscript technically sound, and do the data support the conclusions?

Reviewer #1: Yes

Reviewer #2: Partly

3. Has the statistical analysis been performed appropriately and rigorously? 

Reviewer #1: Yes

Reviewer #2: No

4. Have the authors made all data underlying the findings in their manuscript fully available?

Reviewer #1: Yes

Reviewer #2: Yes

5. Is the manuscript presented in an intelligible fashion and written in standard English?

Reviewer #1: Yes

Reviewer #2: No

6. Review Comments to the Author

Reviewer #1: Journal: PLOS ONE

Article title: Determining the drivers of global innovation under COVID-19:An FSQCA approach

Manuscript ID: PONE-D-23-25968R1

Dear Author (s);

Dear Editor,

The manuscript has been revised for better interpretations according to the suggestions of the Reviewer(s), by including the informations required.

The auhor(s) change the interpretations, results, methodology and conclusions accordingly, and therefore, the paper is much improved now. The author(s) reduces considerabilly the article, references and diversify the articles cited.

I recommend that this article to be published in PLOS ONE.

Congratulations!

Reviewer #2: Dear Authors,

Many thanks for your effort. Your manuscript has come to a far better level through your hard work. Congratulations. I have read your work carefully and provided my observations for the second round below.

• Why the epidemic period is focused? Why do we consider it "epidemic" in 2021? It was pandemic in 2021.

• Abbreviations should be checked. Deleted parts has gone with the long forms of them.

• Assertive hypothetical arguments with determinism still stay (e.g. comments about past innovation indexes)

• Spelling issues still exist (e.g. entrie)

• There are problems on word selection in many cases.

• Deterministic and assertive arguments stay (e.g. “The relationship between income and innovation output is not simply proportional.”; “In the past, the complexity of causality between national innovation input and output had not been considered.”; “ITT is the main motivation for economic activities…….”; “Optimizing the business environment has a positive impact on effective corporate governance, business scope, competitive advantage and innovation ability”)

• At the end of Introduction section, readers expect the aim of the study not implications. Robustness of the method should be discussed in the Methods section careful not in the introduction. “Do we propose a method?” or “Do we propose hypotheses regarding causality between variables?”

• Idea flows are still problematic (e.g. At the national level, the gathering of knowledge workers leads to a region's emphasis on investment in education. The historically good positive cooperation among industry, universities, and -researchers has induced a number of research institutions to serve enterprises by increase their levels of R&D investment. Making it easier for innovators to take advantage of profit from their innovations through intellectual property protection. The level of motivation of innovators to generate new knowledge is insufficient, and the patent system provides a way of increasing the level of such motivations. This is located at the end of “2.2 Construction of the research framework.” Readers expect hypotheses at the end of this section.

• Why shall we accept this argument? (As the GII 2021 is a newly published measurement result, it lacks external and theoretical standards.)

• There are problems in language and logical flow of the ideas and argument. Text should be neatly organized.

• Providing reference does not give you the right to use deterministic and assertive arguments. We are studying social science and all findings should be considered probabilistic. To be on the safe side it is always better to use elements such as “most of …”, “many …..”, “can be /is considered”

• Theoretical ideation part is confusing. I am not convinced about the necessity of the topic and method. “How the gap in the literature is located” and “how it is filled” aspects should be clearly stated combined with the definition of the problem and research question. At end of each theorization there should be clearly stated hypothesis articulating the expected causality between variables and/or combination of variables. Theorizations and arguments proposed by the authprs are not satisfactory.

• We cannot simply blame traditional regression analyses in finding causality. There are ways to test causality using the theoretical bases of regression. Better to use term “linear regression”

• I am not sure if the data set belongs to 2021 or 2020. This is info is not shared.

• For example, in Table 2 China is mentioned. Why are other nations not mentioned?

• Track changes mode version is ok. However, there are several confusing parts. I would like to see the clear form of the latest version corrected following my recommendations.

• The method employed is a robust and promising for cross-sectional data sets in social sciences. However, there should be more explanation about the method. "What are the theoretical bases?" "How these figures are calculated?" Is there a software used for them? "Are they calculated manually?". Besides, implications could be better stated and discussed.

• To conclude, in the first round, I listed many problematic parts as many as I can. Unfortunately, I observe that many of them are not addressed. I try to mention my observation again in this round. The study is not ready for being accepted to be published. However, in terms of methods and the originality of the topic, it deserves to be elaborated more by the authors. I hope I can contribute as a viewer to bring the study to required global standards of academia and PLOS ONE.

Best regards.

7. PLOS authors have the option to publish the peer review history of their article (what does this mean?). If published, this will include your full peer review and any attached files.

Reviewer #1: No

Reviewer #2: No

---

## [Author Response · Author response to Decision Letter 1]

17 Nov 2023

Dear Editors and Reviewers:

Thank you to the editor and reviewers for their suggestions and comments. The comments really helped us see the big picture with the introduction.

Reviewer 1

Thank you very much for agreeing to publish my article in Plos One.

Reviewer 2

Thanks for your encouragement and suggestions.

Comments

• Why the epidemic period is focused? Why do we consider it "epidemic" in 2021? It was pandemic in 2021.

Response: Because the epidemic occurred at the end of 2019, 2020 was the initial stage of the epidemic, which mainly had a great impact on people's life safety, and 2021 was the most serious year, which began to have a major impact on all aspects of the country.

• Abbreviations should be checked. Deleted parts has gone with the long forms of them.

• Spelling issues still exist (e.g. entrie).

• There are problems on word selection in many cases.

Response: We checked the spelling and word selection carefully and made some corrections.

• Assertive hypothetical arguments with determinism still stay (e.g. comments about past innovation indexes).

• Deterministic and assertive arguments stay (e.g. “The relationship between income and innovation output is not simply proportional.”; “In the past, the complexity of causality between national innovation input and output had not been considered.”; “ITT is the main motivation for economic activities…….”; “Optimizing the business environment has a positive impact on effective corporate governance, business scope, competitive advantage and innovation ability”).

• Providing reference does not give you the right to use deterministic and assertive arguments. We are studying social science and all findings should be considered probabilistic. To be on the safe side it is always better to use elements such as “most of …”, “many …..”, “can be /is considered”.

Response: We have perfected the decisive and propositional arguments by collecting literature, and in order to pursue accuracy, we try to use words such as "most …", "many …" or "can/be considered".

• At the end of Introduction section, readers expect the aim of the study not implications. Robustness of the method should be discussed in the Methods section careful not in the introduction. “Do we propose a method?” or “Do we propose hypotheses regarding causality between variables?”

Response: We have modified the introduction to clarify the purpose of the study expected by readers, deleted the robustness of the method, and discussed it in the method part, being careful not to discuss it in the introduction. Like following:

1. INTRODUCTION 

In 2021, an epidemic spread worldwide and altered international social and economic structures. The World Health Organization named it COVID-19. How to manage the virus and when and how to open the economy again have become global issues. In response, governments were faced with the challenges of high-risk research and development (R&D), insufficient investment in innovation caused by the resulting economic downturn, and intensified trade competition due to intercountry control. To address these challenges, it is necessary to rationally allocate resources to promote innovation according to their own national conditions.

Different support systems and cooperative organizations related to industries form an innovative ecosystem that is both interdependent and symbiotic. The relationship between countries is heterogeneously synergetic rather than competitively oppositionist. Actually, in view of the global innovation problem, academic circles have made many beneficial explorations. Many scholars have analyzed mainly the characteristics of innovation and development in a single country or region and made comparisons between these countries or regions. However, most of the above studies explore the independent effects of single variables in accordance with deductive logic, ignoring the combined effect of conditions in the innovation ecosystem. Affected by the epidemic, the global investment policy environment is also full of uncertainties, it is not clear whether the deficiency of a single institutional element hinders innovation and how the conditional configuration systematically affects national innovation. Traditional regression analysis, which simply links income with innovation, may faces practical problems. Since the start of the COVID-19 pandemic, many researches on how to improve global innovation had remained in the stage of theoretical construction, focusing only on the independent role of different factors at the levels of science, technology and organization, which limits the understanding of the synergistic pairing effect of multiple elements of innovation ecosystems to the difference in national innovation output. Based on the configuration perspective, this paper analyzes in-depth whether the configurations and single conditions that drive high innovation cause the bottleneck of national innovation.

The special innovation demand brought about by COVID-19 and the artificially created R&D barriers between countries in response to it have become the constraints of innovation in all countries. Actively responding to the epidemic influence and identifying experiences and lessons from it can provide an empirical basis for countries to complement each other's advantages, innovate collaboratively and overcome difficulties together when a pandemic occurs again. Therefore, against the background of COVID-19, based on the configuration perspective, combined with NCA and FSQCA, this study analyzes the necessary and sufficient causality and identifies the driving path for achieving high innovation.

3.1 The essentiality of using FSQCA and NCA 

Several tenets suggest moving beyond multiple regression analysis to thinking and using algorithms.

First, based on correlation assumes that the variables are independent, the regression analysis method is suitable for exploring the net effect of a single explanatory variable on the explained variable. While FSQCA focuses on analyzing the multiple concurrent causal relationships formed by different combinations of causes and conditions, we can reveal the veil of how the causes and conditions interact together to affect the innovation outcomes (Pappas & Woodside,2021). The FSQCA method begins with holism, proceeds through an analysis of the complex causal relationships among many factors, and finally enables a cross-case comparative analysis to explore which configurations of conditional elements cause the expected results and which fail to lead to the expected result and result in other causal complexity problems. National innovation drives the combination of various factors to form different conditional configurations, and the complex influence on high national innovation output represents this type of problem, so the FSQCA method is especially suitable for this research. 

Second, unlike correlation analysis, consistency is a test for sufficiency and not a test for sufficiency and necessity and FSQCA applies Boolean algebra to overcome the weakness of regression analysis and keeps the complex variable relationship aligned with further information about the research object, which mitigates the issue of missing variable deviation. The traditional quantitative method seeks to obtain the optimal solution of the result, while the qualitative comparative analysis method believes that the configuration leading to the result has equivalence, that is, the combination of multiple different conditions will produce the same result (Farivar et al.,2021). FSQCA can identify the sufficiency and necessity of the causes and conditions that cause the results, as well as the complementarity/substitution among different causes and conditions (Woodside,2013), and further deepen the understanding of different types of innovation input to innovation output.

Finally, the necessary condition refers to the condition needed for a specific result to occur. If this condition does not exist, then the corresponding result cannot be produced. NCA quantitatively demonstrates the antecedent level necessary to achieve a certain level of outcome variables by analyzing the effect size and bottleneck level of the antecedents (%).

• Idea flows are still problematic (e.g. At the national level, the gathering of knowledge workers leads to a region's emphasis on investment in education. The historically good positive cooperation among industry, universities, and -researchers has induced a number of research institutions to serve enterprises by increase their levels of R&D investment. Making it easier for innovators to take advantage of profit from their innovations through intellectual property protection. The level of motivation of innovators to generate new knowledge is insufficient, and the patent system provides a way of increasing the level of such motivations. This is located at the end of “2.2 Construction of the research framework.” Readers expect hypotheses at the end of this section.

Response: According to the requirements of the reviewers, we added some hypotheses leading to national innovation at the end of "2.2 Construction of Research Framework" to make readers better understand. Like following:

Therefore, we assume that ITT, HCR, IFT, MS and BS in the GII structure will have different degrees of influence on national innovation. 

• Theoretical ideation part is confusing. I am not convinced about the necessity of the topic and method. “How the gap in the literature is located” and “how it is filled” aspects should be clearly stated combined with the definition of the problem and research question. At end of each theorization there should be clearly stated hypothesis articulating the expected causality between variables and/or combination of variables. Theorizations and arguments proposed by the authprs are not satisfactory.

Response: We have specially added a part in 3.Research methods to explain the necessity of the method. Like following:

3.1 The essentiality of using FSQCA and NCA 

Several tenets suggest moving beyond multiple regression analysis to thinking and using algorithms.

First, based on correlation assumes that the variables are independent, the regression analysis method is suitable for exploring the net effect of a single explanatory variable on the explained variable. While FSQCA focuses on analyzing the multiple concurrent causal relationships formed by different combinations of causes and conditions, we can reveal the veil of how the causes and conditions interact together to affect the innovation outcomes (Pappas & Woodside,2021). The FSQCA method begins with holism, proceeds through an analysis of the complex causal relationships among many factors, and finally enables a cross-case comparative analysis to explore which configurations of conditional elements cause the expected results and which fail to lead to the expected result and result in other causal complexity problems. National innovation drives the combination of various factors to form different conditional configurations, and the complex influence on high national innovation output represents this type of problem, so the FSQCA method is especially suitable for this research. 

Second, unlike correlation analysis, consistency is a test for sufficiency and not a test for sufficiency and necessity and FSQCA applies Boolean algebra to overcome the weakness of regression analysis and keeps the complex variable relationship aligned with further information about the research object, which mitigates the issue of missing variable deviation. The traditional quantitative method seeks to obtain the optimal solution of the result, while the qualitative comparative analysis method believes that the configuration leading to the result has equivalence, that is, the combination of multiple different conditions will produce the same result (Farivar et al.,2021). FSQCA can identify the sufficiency and necessity of the causes and conditions that cause the results, as well as the complementarity/substitution among different causes and conditions (Woodside,2013), and further deepen the understanding of different types of innovation input to innovation output.

Finally, the necessary condition refers to the condition needed for a specific result to occur. If this condition does not exist, then the corresponding result cannot be produced. NCA quantitatively demonstrates the antecedent level necessary to achieve a certain level of outcome variables by analyzing the effect size and bottleneck level of the antecedents (%).

• We cannot simply blame traditional regression analyses in finding causality. There are ways to test causality using the theoretical bases of regression. Better to use term “linear regression”.

Response: When looking for causality, we don't simply blame the traditional regression analysis, but explain its shortcomings and the advantages of our method. We also replaced "regression analysis" with the term "linear regression" according to the requirements of reviewers.

• I am not sure if the data set belongs to 2021 or 2020. This is info is not shared.

Response: In the article 3.3Date sources, we thought it explained our data source. Like following:

3.3 Data sources.

To demonstrate the technological progress and innovation capability of different countries more intuitively, Cornell University, the Institut Européen d’Administration des Affaires (INSEAD) and the World Intellectual Property Organization (WIPO) jointly published the Global Innovation Index (GII) 2021 edition, which provided indicators with which to measure innovation performance and rank the innovation ecosystems of 132 economies around the world, as 2021 was the year in which the whole world had to jointly deal with COVID-19 (Dutta et al., 2021).The country samples used in this analysis are the same as those referenced in the GII 2021 published in September 2021 (Cornell University, INSEAD, & WIPO, 2021)that reported the innovation of 132 countries worldwide. The choice of each group of countries by income (high-income and low-income countries) is determined using the World Bank income group classification (Cornell University, INSEAD, & WIPO, 2021). 

• The method employed is a robust and promising for cross-sectional data sets in social sciences. However, there should be more explanation about the method. "What are the theoretical bases?" "How these figures are calculated?" Is there a software used for them? "Are they calculated manually?". Besides, implications could be better stated and discussed.

Response: In view of our data analysis software, we have carried on the concrete explanation in the article. Like following: NCA is a method to analyze data by using R software.

Once again, thank you very much for your comments and suggestions.

Thank you and best regards.

Yours sincerely,

Zhenxing Gong and Wag Yue.

---

## [Editor Report · Decision Letter 2]

21 Nov 2023

Determining the drivers of global innovation under COVID-19:An FSQCA approach

PONE-D-23-25968R2

Dear Dr. Yue Wang,

We’re pleased to inform you that your manuscript has been judged scientifically suitable for publication and will be formally accepted for publication once it meets all outstanding technical requirements.

Kind regards,

Shazia Rehman, Ph.D.

Academic Editor

PLOS ONE
---

## [Editor Report · Acceptance letter]

21 Dec 2023

PONE-D-23-25968R2 

PLOS ONE

Dear Dr. Wang, 

I'm pleased to inform you that your manuscript has been deemed suitable for publication in PLOS ONE. Congratulations! Your manuscript is now being handed over to our production team.

Kind regards, 

on behalf of

Dr. Shazia Rehman 

Academic Editor

PLOS ONE